# Optimized phylogenetic clustering of HIV-1 sequence data for public health applications

**Connor Chato**[1], **Yi Feng**[2], **Yuhua Ruan**[2], **Hui Xing**[2], **Joshua Herbeck**[3], **Marcia Kalish**[4], **Art F. Y. Poon**[1]*

**1** Department of Pathology and Laboratory Medicine, Western University, London, Canada, **2** Division of Virology and Immunology, National Center for AIDS/STD Control and Prevention (NCAIDS), Chinese Center for Disease Control and Prevention (China-CDC), Beijing, China, **3** Department of Global Health, University of Washington, Seattle, Washington, United States of America, **4** Department of Medicine, Vanderbilt University, Nashville, Tennessee, United States of America

* apoon42@uwo.ca

**Data Availability Statement:** The purpose of this study is to develop a new statistical methodology. The associated source code has been made publicly available at https://github.com/PoonLab/tn. Anonymized HIV-1 sequence data from northern

## Abstract

Clusters of genetically similar infections suggest rapid transmission and may indicate priorities for public health action or reveal underlying epidemiological processes. However, clusters often require user-defined thresholds and are sensitive to non-epidemiological factors, such as non-random sampling. Consequently the ideal threshold for public health applications varies substantially across settings. Here, we show a method which selects optimal thresholds for phylogenetic (subset tree) clustering based on population. We evaluated this method on HIV-1 *pol* datasets ($n = 14, 221$ sequences) from four sites in USA (Tennessee, Washington), Canada (Northern Alberta) and China (Beijing). Clusters were defined by tips descending from an ancestral node (with a minimum bootstrap support of 95%) through a series of branches, each with a length below a given threshold. Next, we used *pplacer* to graft new cases to the fixed tree by maximum likelihood. We evaluated the effect of varying branch-length thresholds on cluster growth as a count outcome by fitting two Poisson regression models: a null model that predicts growth from cluster size, and an alternative model that includes mean collection date as an additional covariate. The alternative model was favoured by AIC across most thresholds, with optimal (greatest difference in AIC) thresholds ranging 0.007–0.013 across sites. The range of optimal thresholds was more variable when re-sampling 80% of the data by location (IQR $0.008 - 0.016$, $n = 100$ replicates). Our results use prospective phylogenetic cluster growth and suggest that there is more variation in effective thresholds for public health than those typically used in clustering studies.

## Author summary

A genetic cluster of virus infections is a group of DNA or RNA sequences that are much more similar to each other than they are to other infections from the same population of hosts. These clusters can reveal where virus transmission has been occurring the most rapidly, which can provide useful information for a public health response. Genetic clusters

Alberta, Canada, were retrieved from Genbank (accession numbers KU189996-KU191050). In addition, we provide anonymized and randomized sequence alignments of the three other data sets at the above URL. Random permutation of columns for these alignments destroys all resemblance to the original HIV-1 sequences, maintaining privacy while preserving the phylogenetic relationships. Requests for access to additional data may submitted to the following contacts or web forms: the Seattle/King County Public Health Agency (HIVepi@kingcounty.gov); the Vanderbilt Comprehensive Care Clinic (VCCC, https://www.vumc.org/tncfar/clinical-sciences-core); and the Chinese Center for Disease Control and Prevention (fengyi@chinaaids.cn or office606@chinaaids.cn).

**Funding:** This study was supported by a grant from the Canadian Institutes of Health Research (PJT-156178) to AFYP and by an Administrative Supplement to a Center Core Grant from the Tennessee Center for AIDS Research (P30-AI110527) to MK. The funders had no role in study design, data collection and analysis, decision to publish, or preparation of the manuscript.

**Competing interests:** The authors have declared that no competing interests exist.

are often built by reconstructing a phylogeny—a tree-based model of how the sequences are related by common ancestors—and locating distinct parts of the tree with short branches. However, there are no objective, general-purpose criteria for deciding which parts of a tree constitute clusters, and there are an unlimited number of ways to partition a tree into clusters. In this study, we develop a computational method to determine the best clustering criteria based on our ability to predict where the next infections will occur. We apply this method to anonymized HIV-1 sequence data sets from Canada, the United States, and China, to characterize the sensitivity of clustering criteria to different risk populations and sampling contexts. Our results indicate that the clustering criteria typically used for phylogenetic studies of HIV-1 are not optimal for public health applications.

## Introduction

Identifying clusters of infections with shared characteristics that imply a common origin is a fundamental goal for epidemiological research and surveillance. This can be used to inform regional lockdown protocols and identify subpopulations for prioritized allocation of public health resources [1, 2]. Conventionally, clusters are identified by infections that are sampled within a relatively short time frame (temporal clustering), in association with a defined space (spatial clustering), or both. However, molecular sequence data have been used increasingly to cluster infections by their genetic similarity, which can provide a complementary or surrogate measure of their spatial or temporal proximity. There are now many examples of genetic clustering applied to RNA viruses, including Ebola virus [3, 4], human immunodeficiency virus type 1 (HIV-1) [5–7], hepatitis C virus [8], and coronaviruses associated with Middle Eastern respiratory syndrome (MERS-CoV) [9] and the 2003 outbreak of severe acute respiratory syndrome (SARS-CoV) [10]. The rapid evolution of many RNA viruses favours the use of genetic clustering because mutational differences can accumulate between infections in a matter of weeks or months [11, 12]. When transmission occurs on a similar time scale, a molecular phylogeny reconstructed from infections of an RNA virus will be shaped in part by its transmission history [13]. In this context, a phylogeny is a tree-based model of how infections are descended from their common ancestors.

HIV-1 has been particularly targeted for applications of genetic clustering. This is driven not only by HIV-1's rapid evolution and global impact, but also by the availability of large sequence databases in many clinical settings from routine screening for drug resistance mutations. For instance, continual updates to sequence databases make it feasible to monitor the emergence and growth of genetic clusters over time [5, 14, 15]. Clustering methods have been used retrospectively to characterize populations associated with elevated rates of HIV-1 infection [7, 16–19], to identify potentially linked co-infections of HIV and hepatitis C virus [8, 20], superinfection by multiple HIV-1 subtypes [21] and transmitted drug resistance [22]. Clustering has also been proposed to support HIV-specific prevention methods such as pre-exposure prophylaxis (PrEP), which require a precise understanding of high-risk populations to optimize the distribution of public health resources [23, 24]. Finally, the lack of an effective vaccine demands a continuous assessment of priority populations for testing and antiretroviral treatment [25].

Over the last decade, there has been a growing diversity of genetic clustering methods, many of which were specifically designed and validated on HIV-1 sequence data [6, 26–31]. These methods can be broadly categorized by whether or not they require the reconstruction of a molecular phylogeny [32]. For example, Cluster Picker [26] is currently among the most

widely-used genetic clustering methods for HIV-1 based on numbers of citations in the literature. Cluster Picker defines each cluster as a subtree in the phylogeny. A subtree is a portion of the phylogenetic tree that consists of an ancestral node and all of its descendants; in evolutionary terminology, a subtree is a monophyletic group. Cluster Picker searches for subtrees where (1) the total length of branches between any pair of tips in the subtree ($d$) is always below a threshold $d_{max}$, and (2) the bootstrap support associated with the ancestral node exceeds a threshold $b_{min}$. A bootstrap support value ($b$) is a measure of reproducibility—how often we expect to reconstruct a node ancestral to exactly the same set of tip labels from a hypothetical new data set of equivalent dimensions to the original data [33]. In practice, $d_{max}$ and $b_{min}$ are typically set to values in the range of 0.01 to 0.05 expected nucleotide substitutions per site and 90% to 95%, respectively [19, 26, 34, 35].

Clustering methods that do not reconstruct a phylogeny are also in widespread use for HIV-1 [15, 36–38]. For example, HIV-TRACE [6] employs a genetic distance [39] that reduces two sequences to a number quantifying the extent of their evolutionary divergence. Clusters are assembled from pairs of sequences whose distances fall below some predefined threshold. The advantages of distance-based clustering is that pairwise distances are rapid to compute and yield immutable quantities; these distances do not change with the addition of sequences to the database. In contrast, reconstructing a phylogeny with additional sequence data can change the branch lengths and bootstrap support values associated with previously-defined clusters [35]. Consequently, the use of clustering for continuous monitoring of an HIV-1 sequence database (*i.e.*, to track the growth of clusters) has tended to focus on distance clustering methods [15, 38]. Tracking cluster growth can provide more informative indicators for public health decisions. For instance, large clusters tend to emphasize historical outbreaks that are no longer active [38, 40].

Nevertheless, phylogenetic clustering remains more prevalent in the infectious disease literature [41]. Clusters generated from pairwise distances tend to have a high density of connections (edges) between cases, resulting in swarms of connections that are difficult to interpret, an instance of the 'hairball' problem that plagues applications of networks to social and biological systems [42, 43]. In contrast, phylogenetic clusters are generally more interpretable as trees that are shaped in part by the underlying transmission history. Both genetic distance and phylogenetic clustering methods require users to select one or more threshold parameters. Clustering results can vary substantially with different methods and clustering thresholds [44]. There are no universal guidelines for configuring a phylogenetic clustering method for applications in public health and molecular epidemiology. In the absence of generic data-driven methods to select optimal thresholds, many users have resorted to default settings and/or emerging conventions in the literature [41, 44]. However, appropriate thresholds can vary substantially among databases and populations, due to differences in prevalence of infection, extent of sampling, and heterogeneity in rates of transmission and diagnosis [32, 45, 46].

Relaxing clustering thresholds tend to yield larger clusters [18, 36, 37]. Several studies [30, 32, 34, 47] have noted that larger clusters tend to have more information for predicting the location of the next cases. However, this favours relaxing the clustering thresholds until all sequences merge to a single giant cluster, and the appearance of the next cases in that cluster is a trivial outcome [48]. An analogous information-bias tradeoff has been studied extensively in spatial statistics, where it is known as the modifiable area unit problem [49, 50]. Thus, we have adapted a strategy that was proposed to study the distribution of mortality rates for varying levels of administrative districts in Tokyo [51] (*e.g.*, ward, town, village). This method compares the AICs of two Poisson regression models for varying numbers of clusters. Although AICs for one model cannot be directly compared for different clusterings (since the data are

being changed), the clustering that maximizes the difference in AICs between models (ΔAIC) minimizes information loss with the addition of model parameters [52, 53].

In previous work [48], we developed a statistical framework based on ΔAIC to select the optimal threshold for distance-based clustering. This optimum is based on one's ability to predict the distribution of the next cases of infection among existing clusters. Here, we extend this framework to enable users to calibrate phylogenetic clustering methods to a specific population database. This is not a trivial task because we must accommodate the effect of new data on the shape of the phylogenetic tree, such that new cases may retroactively change previous clusters. We adapt a maximum likelihood method (*pplacer* [54]) to graft new sequences onto a pre-existing phylogeny. Next, we fit predictive models of cluster growth based on the placement of new cases on the tree. To optimize the phylogenetic clustering method to a given data set, we evaluate these models for a range of branch-length and bootstrap support thresholds. We assess the performance of our method on HIV-1 sequence data sets from two regions of the United States (Tennessee [55] and Washington state [56]), the northern region of Alberta in Canada [57], and Beijing, China [58].

## Methods

### Data collection

This study was performed on alignments of anonymized HIV-1 *pol* sequences, where each sequence uniquely represented a host individual. Aligned sequence data were obtained from different locations: Kings County, Washington, USA ($n = 6815$) [56]; Middle Tennessee, USA ($n = 2779$) [55]; Beijing, China ($n = 3964$) [58]; and northern Alberta, Canada ($n = 1054$) [57]. The first three data sets were acquired with special permission from the Seattle/King County Public Health Agency [56, 59], the Vanderbilt Comprehensive Care Clinic [14] and the Chinese Center for Disease Control and Prevention, respectively. The fourth data set is publicly available in Genbank (accession numbers KU189996—KU191050) [57] and we used a custom R script to extract collection dates and HIV-1 subtype classifications from sequence headers. Data sets were filtered to remove any sequences with over 5% ambiguous sites and sequences that were over 15% incomplete. We also manually examined and trimmed the sequence alignments for the Washington and Tennessee data sets due to a relatively high proportion of gaps and ambiguous base calls, removing a total of 52 and 163 nt from the overall alignment lengths, respectively. To make the results of our analysis as consistent as possible for the respective data contributors, we refrained from further modifications to the alignments.

All sequences were associated with years of sample collection as metadata. For each data set, sequences were partitioned by date of sample collection, with sequences in the most recent year comprising the 'incident' subset, and the remaining sequences assigned to the 'background' subset. In addition, year of HIV diagnosis was available for all sequences in the Washington data set, and for a subset of sequences in the Tennessee data set (Table 1). For these cases, we repeated our analysis on partitions of the sequence data by year of diagnosis. The composition of the final sequence data sets are summarized in Table 1.

### Phylogenetic analysis

Maximum likelihood phylogenies were constructed for each set of background sequences using IQ-TREE version 1.6.12 [61] with a generalized time-reversible (GTR) model of evolution, gamma-distributed rate variation among sites, and 1,000 parametric bootstraps. In addition, 100 random sub-samples were drawn without replacement from background sequences, each containing 80% of the full sample population. FastTree2 version 2.1.10 [62] was then used to construct a separate tree for each sub-sample, using default run parameters and 100

**Table 1. Summary of sequence data characteristics.** Length is the median length of nucleotide (nt) sequences. HXB2 coords = reference nucleotide coordinates in the HXB2 genome (Genbank accession K03455). Year type: sequences are annotated with year of sample collection, and in some cases date of HIV diagnosis. *N* = Total sample size, including both old and new sequences. Incid = number of sequences in 'incident' subset (most recent year). Subtype classifications were derived from the original data sources, when available, or generated *de novo* with SCUEAL [60].

| Location | Length (nt) | HXB2 coords | Year type | *N* | Incid | Date range | Subtypes |
|---|---|---|---|---|---|---|---|
| Washington, USA | 985 | 2,256–3,240 | diagnosis | 6,583 | 253 | 1982–2019 | B (89%), C (4%), A1 (1%), other (10%) |
| | | | collection | 6,583 | 253 | 1999–2019 | |
| Alberta, Canada | 1,017 | 2,253–3,269 | collection | 1,051 | 155 | 2007–2013 | B (77%), C (20%), A1 (3%) |
| Tennessee, USA | 1,296 | 2,253–3,548 | collection | 2,741 | 162 | 2001–2015 | B (93%), C (2%), other (5%) |
| | | | diagnosis | 2,338 | 129 | 1977–2011 | |
| Beijing, China | 1,004 | 2,273–3,276 | collection | 3,916 | 1,196 | 2005–2015 | 01AE (50%), 07BC (25%), B (19%), other (6%) |

bootstraps. All trees were midpoint-rooted using the *phytools* [63] package in R. Polytomies were resolved arbitrarily using the *multi2di* function in the R package *ape* [64]—this was only necessary for sub-samples drawn from the Washington data set. Patristic distances (tip-to-tip branch lengths) were calculated using the R package *ape* function *cophenetic.phylo* and the pairwise distances were calculated using a Tamura-Nei [39] distance calculator (TN93, https://github.com/veg/tn93).

To update these phylogenies with incident sequences, we used the software package *pplacer* version 1.1 [54] to graft the sequences onto a tree by maximum likelihood. In many cases, the placement of a sequence varied among bootstrap replicates. The confidence levels among alternative placement queries ('pqueries') was quantified by likelihood weights. Each pquery is associated with a terminal branch between the incident sequence and the ancestral node *A* where it bisects the tree, as well as a 'pendant' branch between *A* and its descendant in the original tree (Fig 1). We used the *sing* subcommand in the *guppy* program within *pplacer* to generate trees with the different pqueries [65], ensuring that graft locations were only proposed on branches that existed in the original tree.

## Cluster definition

Taking a phylogenetic tree of background sequences as input, we implemented the following set of rules in R to assign incident sequences to a set of clusters in the tree:

1. A cluster is a subset of branches and nodes within a binary tree rooted on ancestral node *A* and containing at least one terminal node.

2. By default, a single terminal node constitutes a cluster of one.

3. Each terminal node in a cluster must descend from the root node of the cluster through a path of branches whose lengths each fall below $d_{max}$.

4. The ancestral node *A* must have a bootstrap confidence equal to or exceeding some threshold $b_{min}$.

5. Any node *N* and its associated parent branch may only belong to the largest possible cluster containing *N*.

By these rules, a cluster can be as small as a single sequence, but can also be represented by either monophyletic or paraphyletic groups. The latter allows relatively divergent sequences to be separated from an otherwise closely related group, leaving the latter intact. Clusters with the same number of sequences can also vary by how many branches and internal nodes are

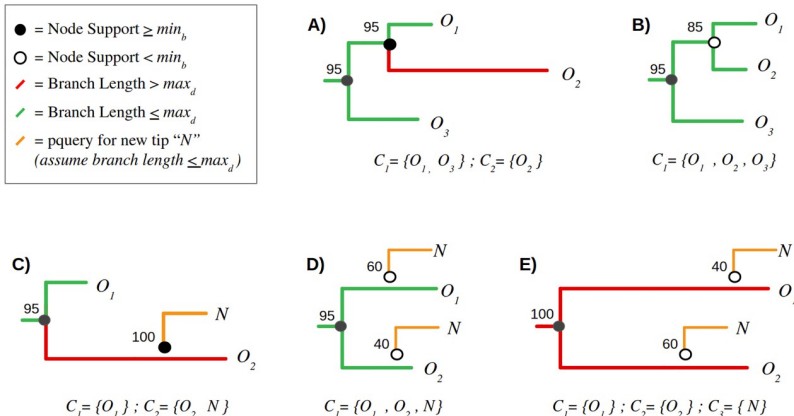

**Fig 1. Examples of clustering definition and growth criteria.** Subtree (A) represents a paraphyletic cluster ($C_1$) of two background (old) sequences, $O_1$ and $O_3$, excluding a third sequence $O_2$ that is too distance from the root node of $C_1$. Consequently, $O_2$ becomes its own cluster of one, $C_2$. Subtree (B) illustrates a monophyletic cluster where all background sequences meet this criterion. Subtrees (C-E) depict the addition of a new (incident) sequence $N$ to an existing cluster. In (C), the new sequence is added with 100% confidence to a cluster of one background sequence, $O_2$. Conversely, the placement of $N$ in subtrees (D) and (E) is highly uncertain (with bootstrap supports 40% and 60%). For (D), $N$ becomes incorporated into the same cluster irrespective of its placement, so the bootstrap values are irrelevant. In contrast, neither placement of $N$ in subtree (E) meets the clustering criteria due to the resulting branch lengths—as a result, $N$ becomes a new cluster of one, $C_3$.

included because of unresolved nodes (polytomies), although we do not utilize these quantities in our methods. We only count terminal nodes to compute the size of a cluster.

We generated clusters at 41 different branch length thresholds, ranging from $d_{max} = 0$ to 0.04 in steps of 0.001. Clusters were also generated under relaxed ($b_{min} = 0$) and strict ($b_{min} = 0.95$) bootstrap thresholds. Since our results were relatively insensitive to varying $b_{min}$, we limited our analyses to these two settings. For the purpose of clustering, the root node of the tree was assigned a bootstrap support of 0, such that the entire tree could only become subsumed into a single giant cluster at the lowest threshold $b_{min} = 0$.

For comparison, we implemented a similar set of clustering criteria in the program Cluster Picker [26], which is widely used for the phylogenetic clustering analysis of HIV-1 sequences. This required two modifications to the above rules. First, a cluster selected by Cluster Picker must be a monophyletic group (containing all descendants) rooted on an ancestral node with a bootstrap support greater than or equal to a threshold $b_{min}$. Second, all patristic distances in the cluster cannot exceed the threshold $d_{max}$. We evaluated Cluster Picker at a broader range of distance thresholds, from $d_{max} = 0$ to 0.08 in steps of 0.002, and with $b_{min} = 0.95$. To distinguish our method from Cluster Picker, we will herein refer to the former as 'paraphyletic clustering'.

## Simulation

We used *pplacer* to simulate the growth of clusters by the addition of incident sequences onto a fixed tree. This grafting creates new internal nodes and terminal branches and nodes. Next, we re-ran our paraphyletic clustering algorithm on the updated tree. To prevent any case where two clusters merge into one, only terminal or pendant branches descending from a newly created internal node were evaluated for clustering. It is possible for a single sequence to generate multiple proposed locations with varying confidence levels because *pplacer* employs bootstrap resampling. To avoid ambiguous placements, the internal node created by grafting a

new sequence must have a bootstrap support that exceeds the threshold $b_{min}$. If multiple placements with low support exist within the same cluster, we sum the support values to determine whether to assign the sequence to the cluster. We provide several examples of cluster growth in Fig 1 to clarify these definitions.

We also use *pplacer* to measure growth for the monophyletic clustering method in Cluster Picker. Unlike our paraphyletic clustering method, the addition of new tips in Cluster Picker has the potential to separate an existing cluster by introducing a new terminal branch with a length that exceeds $d_{max}$. To prevent this kind of retroactive disruption to the composition of existing clusters, we only considered placements that would not result in a new distance over $d_{max}$ for growth.

## Predictive performance analysis

Following our previous work [48], we used the predictive value of a Poisson regression model to optimize clustering parameters. In other words, we modelled the addition of incident cases to clusters of background sequences as a count outcome, with a cluster-specific rate $\lambda_i$ determined by the original composition of the $i$-th cluster. We propose that this approach is the most consistent with the implicit objectives of public health applications of clustering. We evaluated two Poisson models incorporating different sets of predictor variables. In the first model, cluster growth is predicted solely by the total number of background sequences in the cluster, also known as cluster size. This assumes that every individual background sequence has the same probability of being the closest relative to the next incident sequence. Thus, larger clusters tend to accumulate more new cases simply because they are large.

A second, more complex model incorporates an additional term corresponding to the mean 'age' or recency of sequences in the cluster relative to the current time. Age may be calculated from either dates of sample collection or diagnosis. Written more generally, we let $g(c)$ represent the expected growth of cluster $c$:

$$g(c) = \exp(\beta + \alpha_1 x_1 + \alpha_2 x_2 + \ldots) \tag{1}$$

where $x_i$ represents the $i$-th predictor variable, such as cluster size. Thus, this linear model can be modified to accommodate any number of predictor variables. The unit of observation for this model is a cluster of background sequences. For any model, we used the *glm* function in R to estimate the coefficients $\alpha_i$ and intercept $\beta$.

We used the Akaike information criterion (AIC) [66] to compare the fit of alternative models on the same data. AIC increases both with model prediction error as well as the number of model parameters. To restate our central postulate, the optimal clustering threshold(s) define a partition of the data into clusters that maximizes the difference between the AIC of the null model, $g_0(c) = \exp(\beta + \alpha_1 x_1)$, which uses only cluster size ($x_1$); and the AIC of an alternative model $g_1(c) = \exp(\beta + \alpha_1 x_1 + \alpha_2 x_2)$ that incorporates both cluster size and mean cluster age ($x_2$). We measure this difference by $\Delta AIC = AIC(g_1) - AIC(g_0)$ and determine which combination of clustering thresholds $d_{max}$ and $b_{min}$ minimizes this quantity, *i.e.*, attains the most negative value. At these optimized thresholds, the selection of a more informed model is the least ambiguous. We expect $\Delta AIC$ to approach zero at the most relaxed thresholds, where all background sequences are placed into a single giant cluster [48], such that it is not possible to differentiate between model outcomes. Conversely, at the strictest thresholds every background sequence is assigned to its own cluster of one. The predictive value of any characteristic such as age is diminished by extremely small sample size of each cluster, so we also expect $\Delta AIC$ to approach zero in this scenario.

In addition to the AIC-based approach, we used the R package pROC [67] to generate receiver operator characteristic (ROC) curves for varying cluster size and age thresholds. An ROC curve plots the true positive rate (TPR, sensitivity) and true negative rate (TNR, specificity) of a binary classifier when varying a single tuning parameter. Since our model predicts count-valued outcomes, we dichotomize the results to predict whether a cluster will grow by one or more cases $g(c) > 0$ or not $g(c) = 0$. For example, a true positive corresponds to a cluster that was both predicted and observed to accumulate one or more cases. Furthermore, we calculated the area under the curve (AUC) to provide a more conventional measure of model performance, where an AUC of 1 indicates perfect prediction.

These phylogenetic clustering, simulation and model-fitting methods are released as an R package at https://github.com/PoonLab/tn under the GNU General Public License (GPL) version 3.

## Validation of clusters

To evaluate whether subtree clusters extracted under our optimized criteria actually represent epidemiologically-related infections, *i.e.*, whether the incident sequences are being correctly grafted onto known clusters, we used FAVITES (version 1.2.10) [68] to simulate an epidemic in a network and to sample a transmission tree and simulate sequence evolution. We used the same configuration as a previous study of using phylogenetic clustering to prioritize HIV prevention measures [69]. In brief, FAVITES was configured to generate a random network under the Barabási-Albert [70] preferential attachment model, in which highly connected nodes tend to accumulate more connections. Next, an epidemic was seeded with 3,110 infections in a population of 26,746 individuals, and propagated for 10 simulation-years through the network using the program GEMFsim [71] under the HIV-1 transmission model described by Granich and colleagues [72]. This model was parameterized so that HIV-1 transmission occurred predominantly from untreated individuals at an early stage of infection [69]. Infected individuals are sampled upon initiating ART. To simulate a virus phylogeny from the resulting transmission tree, virus population dynamics within each host individual was simulated under a logistic coalescent model. Finally, HIV-1 sequence evolution was simulated along the virus phylogeny under a generalized time-reversible model of nucleotide substitution that was previously parameterized to actual HIV-1 *pol* sequences [69]. These sequences were labelled by sampling times in simulation time units, which we converted to standard dates.

To validate our clustering method, we used FastTree to reconstruct a maximum likelihood tree from the sequence alignment generated by FAVITES, and used the resulting tree and log files as inputs for our method. Sequences collected in the final year of the simulation were classified as incident cases. In addition, we downsampled the number of sequences to a maximum of 500 per year, so that the sample sizes were more similar to the actual HIV-1 data sets we evaluated in this study. We used *pplacer* to graft incident sequences to the tree, extracted clusters under varying distance thresholds and bootstrap thresholds of 0% and 95%. Next, we generated the ΔAIC profiles to locate the optimal clustering thresholds based on the distribution of incident sequences among clusters. Finally, we compared the distribution of incident sequences among phylogenetic clusters of known sequences to the true transmission history produced by FAVITES, using the R package *igraph* to analyze the latter as a directed graph.

## Results

### Phylogenetic analysis

We reconstructed maximum likelihood phylogenies for alignments of HIV-1 *pol* sequences from each of four different sites (Washington and Tennessee, USA; Alberta, Canada; and

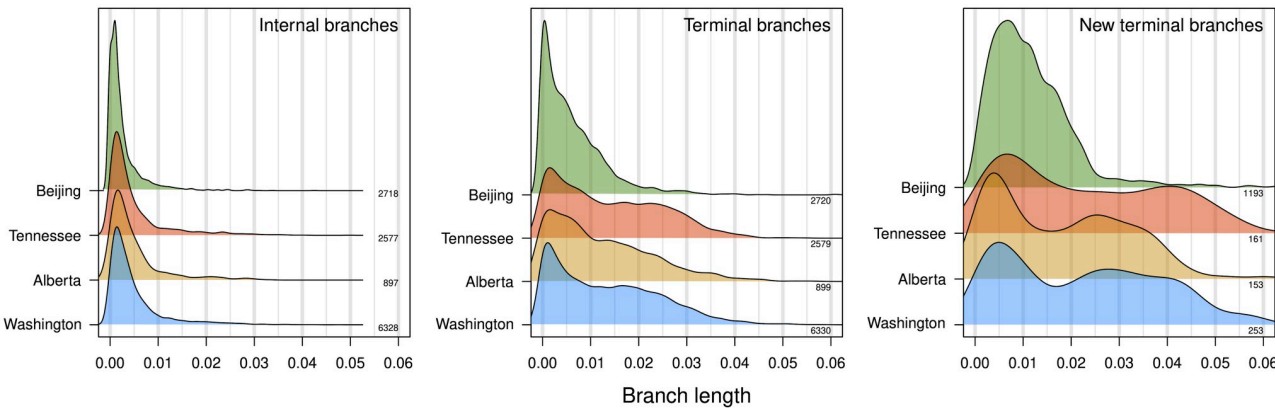

**Fig 2. Distributions of branch lengths in HIV-1 *pol* phylogenies.** Each density summarizes the distribution of branch lengths (measured in units of expected nucleotide substitutions per site) for different locations and types of branches, as indicated in the upper right corner of each plot. We used Gaussian kernel densities with default bandwidths adjusted by factors 1.5, 0.75 and 0.75, respectively. Densities are labeled on the right with the corresponding number of branches. Internal and terminal branches are derived from the phylogeny reconstructed from background sequences. New terminal branches refer to additional branches to incident (new) sequences as placed onto the phylogeny by maximum likelihood (pplacer).

Beijing, China). These alignments excluded all new (incident) sequences that were collected in the most recent year for each site. Fig 2 summarizes the distribution of branch lengths in the resulting trees. Although some of the data sets comprised multiple HIV-1 subtypes, this has a negligible effect on the internal branch length distributions because only a relatively small number of branches separate subtypes. Overall the distributions of internal and terminal branch lengths were broadly similar among locations. Internal branch lengths tended to be substantially shorter than terminal branch lengths (Fig 2), which is typical for HIV-1 among-host phylogenies. In addition, branches created by grafting new (incident) sequences onto the existing trees were significantly longer than terminal branches in the original trees (Wilcoxon test, $P < 10^{-10}$; Fig 2). This trend may be driven by constraints imposed on the original tree when placing new sequences by maximum likelihood [54].

The overall mean patristic distances were 0.102 (Washington), 0.115 (Alberta), 0.086 (Tennessee) and 0.159 (Beijing); the mean Tamura-Nei (TN93) distances ranged from 0.066 to 0.089, and were highly correlated with the patristic distances (Pearson's $r^2$ = 0.995). Differences in mean patristic distances among locations were driven in part by the presence of multiple major HIV-1 subtypes in the Alberta and Beijing phylogenies as distinct groups. For instance, the mean patristic distances within subtypes in the Beijing data set were 0.0630 (CRF01_AE), 0.0564 (CRF15) and 0.0564 (B). For Alberta, the mean patristic distances within subtypes were 0.067 (A1), 0.078 (B) and 0.074 (C). No cases of super-infection were reported in studies associated with these data sets, *e.g.*, [55, 56, 58, 73].

## Predicting cluster growth

For each phylogeny of background sequences (excluding the most recent year), we extracted clusters under varying branch lengths ($d_{max}$ from 0 to 0.04 in increments of 0.001 expected nucleotide substitutions per site) and bootstrap support (0% and 95%) thresholds. We measured cluster growth by the placement of new HIV-1 sequences (from the most recent year) onto the respective tree by maximum likelihood. As implied by the distribution of new terminal branch lengths (Fig 2), relaxing the branch length threshold ($d_{max}$) resulted in higher rates of cluster growth. At the highest $d_{max}$ evaluated in this study (0.04), 81% (Washington), 94%

(Alberta), 77% (Tennessee) and 98% (Beijing) of all new sequences were grafted into existing clusters (S1 Fig). However, this high threshold also tended to collapse the background sequences into a single giant cluster, including sequences from different HIV-1 subtypes. Thus, at $d_{max}$ exceeding 0.04, phylogenetic clusters are no more epidemiologically informative than the classification of the sequences into HIV-1 subtypes [74].

We modelled cluster growth as a Poisson-distributed outcome. Specifically, we evaluated different log-linked models using (1) the numbers of sequences per cluster, *i.e.*, cluster size, or; (2) both cluster size and the mean times associated with sequences in clusters as predictor variables. We use the simplest case (cluster size only) as our null model. Times were based on either dates of sample collection or HIV diagnosis for the respective individuals. For instance, we expect a cluster comprising more recent infections to be more likely to gain new sequences. Fitting these models to a given data set provided two values of the Akaike information criterion (AIC), which measures the fit of the model penalized by the number of parameters. At a given set of thresholds defining clusters, the difference in AIC between these models quantifies the information gain by the addition of mean cluster times. At the most relaxed threshold, all sequences belong to a single cluster, and the addition of predictor variables has no effect on model fit. Conversely, at the strictest thresholds every background sequence is a cluster of one, such that the distribution of new sequences is essentially random with respect to individual characteristics. Thus, the impact of predictor variables on model information is contingent on how we partition the sample population.

The optimal clustering thresholds should resolve the bias-variance tradeoff between overfitting small clusters and underfitting large clusters. Put another way, the optimal thresholds minimize the information loss associated with the addition of one or more predictor variables, relative to the null model [48]. We quantify this information gain by computing the difference in AIC ($\Delta$AIC) between the two models. Fig 3 illustrates the profiles of $\Delta$AIC for the different data sets with respect to varying branch length (distance) thresholds with a fixed 95% bootstrap support threshold. In all four cases, $\Delta$AIC was the most negative at intermediate distance thresholds, which varied slightly among locations: 0.007 (Alberta), 0.008 (Beijing), 0.012 (Tennessee), and 0.013 (Washington). We note that these optimal distance thresholds tend to be shorter than the thresholds often used in the literature (*e.g.*, 0.045) after adjusting for the use of branch lengths (this study) versus tip-to-tip distances (*e.g.*, Cluster Picker). The proportion of new sequences mapped to clusters at these optimal $d_{max}$ thresholds were: 34.6% (Alberta), 40.2% (Beijing), 35.4% (Tennessee), and 39.1% (Washington; S1 Fig).

For a secondary measure of performance, the area under the curve (AUC) was obtained for receiver-operator characteristic (ROC) curves associated with the two Poisson regression models. In this case, cluster growth (using the paraphyletic method with $b_{min} = 0.95$) was reduced to a binary outcome. Put another way, we evaluated our ability to predict which clusters would grow by the addition of one or more new cases. At their respective optimal values of $d_{max}$, we obtained AUC values of 0.87 (Washington), 0.62 (Alberta), 0.76 (Tennessee) and 0.67 (Beijing) for the Poisson model using both cluster size and recency as predictor variables. These AUC values tended to be higher than the values obtained under the null model using only cluster size (S2 Fig). The exception was the Alberta data set, where similar AUC values were obtained from either model. In fact, the null model yielded significantly higher AUC values (paired Wilcoxon signed rank test, $P = 0.01$), and the AUCs were almost identical at the $\Delta$AIC-selected $d_{max}$ (0.629 null against 0.617 alternative). In contrast to our results for $\Delta$AIC (Fig 3), the AUC profiles did not consistently select for an optimal $d_{max}$ threshold, *i.e.*, a global maximum. These results imply that AUC is not a reliable statistic for calibrating phylogenetic clustering methods on the basis of prospective cluster growth.

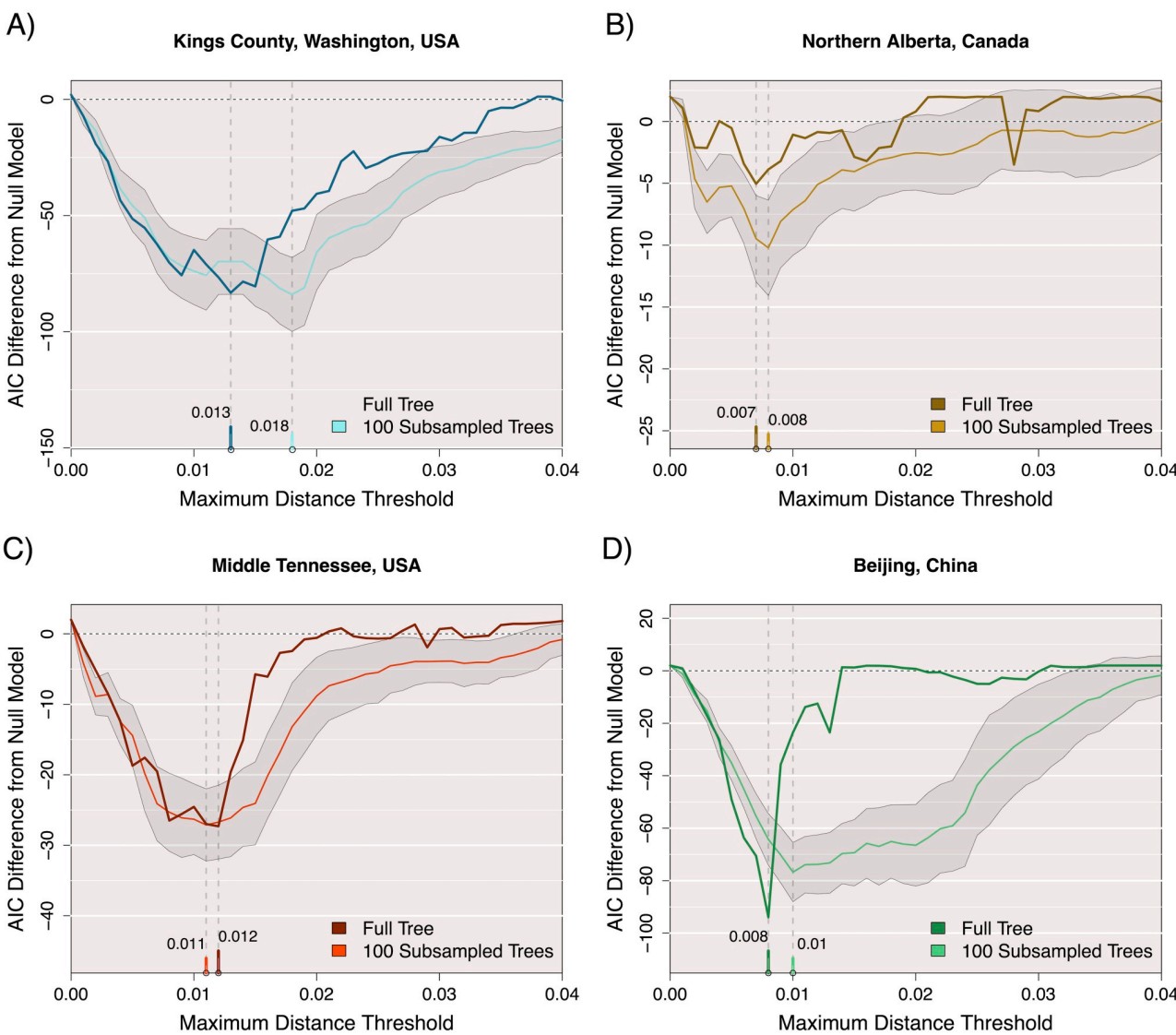

**Fig 3. Difference in AIC between Poisson-linked models of cluster growth.** Clusters and growth are defined at 41 different maximum distance thresholds from 0 to 0.04 with a minimum bootstrap support requirement of 95% for ancestral nodes. The AIC of a null model where size predicts growth is subtracted from the AIC of a proposed model where size and mean time predict growth. The darker colour in each plot corresponds to these AIC results for a maxmimum likelihood tree built from the full set of old sequences, while the lighter colour represents the mean AIC difference obtained by this threshold for 100 approximate likelihood trees built on 80% subsamples of the old sequences without replacement. The shaded area represents 1 standard deviation from the mean AIC difference for subsamples at this threshold. Date of sequence collection was used to measure time for all data sets.

## Sensitivity analysis

To evaluate the sensitivity of these results to sample size and random variation, we sampled 80% of the background sequences at random without replacement to generate 100 replicates, and repeated our analysis for each sample. The distributions of the resulting optimal distance thresholds are summarized in Fig 3 and S3 Fig. The median optimal distance threshold for sub-samples of the Washington data was substantially greater (0.018) than the optimum for the full tree (0.013). This implied that the models required a more relaxed distance threshold to capture similar patterns of cluster growth with reduced sample sizes. However, subsetting

the other data sets did not substantially change the optimal distance thresholds. Sub-sampling the Beijing data set resulted in an unusually broad ΔAIC profile, relative to the full data profile, and the distribution of optima among samples included a distinct 'shoulder' around a distance threshold of 0.02 (S3 Fig). Examining the cumulative plots of new sequences in clusters with increasing $d_{max}$ (S1 Fig), we note that the optimal thresholds tended to coincide with the 'elbow' of the respective curves except for Beijing. This implies that the broader distribution of ΔAIC values associated with this data set may be driven by the atypically high number of incident sequences. In addition, we progressively right-censored the data sets with respect to years of sample collection, such that we evaluated the distribution of incident sequences from earlier years among clusters for a reduced set of background sequences. For the Washington and Tennessee data sets, this right-censoring caused the optimal distance thresholds to drift over time (S4 Fig). Depending on the window of collection dates used to determine clusters, the optimal thresholds ranged from 0.008 to 0.021 for Tennessee and from 0.009 to 0.017 for Washington. This variation is comparable to what we observed from random sub-sampling of the data sets (S3 Fig).

### Effect of bootstrap support

For all data sets, we generated alternative sets of AIC loss results with the minimum bootstrap threshold ($b_{min}$) reduced to 0. The original threshold of $b_{min} = 95\%$ prevented many mid-sized clusters from forming. Specifically, 49% to 74% of internal nodes in each complete tree failed this threshold. This requirement also had a dramatic effect on the growth of singleton and small clusters, as 88%—93% of the new sequence placements had $b < 95\%$, often resulting in clusters growing only through multiple placements. Overall, relaxing $b_{min}$ tended to reduce the loss of information, as reflected by lower values of ΔAIC (Fig 4). This outcome was the most apparent in the Washington data set, which had the largest proportion of internal nodes above $b_{min} = 95\%$. Relaxing $b_{min}$ also tended to induce a shift in the optimal $d_{max}$ as defined by minimizing ΔAIC, although this appeared to be a stochastic outcome. In contrast, relaxing $b_{min}$ for the Beijing data set resulted in higher values of ΔAIC in the neighbourhood of the optimal $d_{max}$ threshold. Again we attribute this difference to the atypically high number of incident cases for this data set. In sum, our results imply that relaxing the threshold $b_{min}$ tends to confer a prediction advantage by incorporating a greater number of clusters into the analysis, even though many of those clusters would not be consistently reproducible with new data.

### Diagnosis versus collection dates

For the Washington and Tennessee data sets, we generated an additional set of ΔAIC profiles cluster ages computed from the available dates of HIV diagnosis instead of sample collection dates. Given that dates of HIV diagnosis tend to be closer to the actual dates of infection, we expect these metadata to be more useful for predicting the distribution of new cases. Indeed, using the diagnosis dates resulted in substantially lower ΔAIC values in either data set, irrespective of bootstrap thresholding (Fig 4). The optimal $d_{max}$ as determined by ΔAIC was invariant to using either set of dates for the Washington data set. On the other hand, we obtained slightly different optima for Tennessee, most likely because diagnostic dates were only available for a subset of incident cases in this data set (80%, Table 1).

### Monophyletic clustering

Finally, we generated another set of ΔAIC profiles using clusters that were constrained to be monophyletic, *i.e.*, comprising all descendants of the internal node. This second clustering method is more similar to the phylogenetic clustering methods used in the molecular

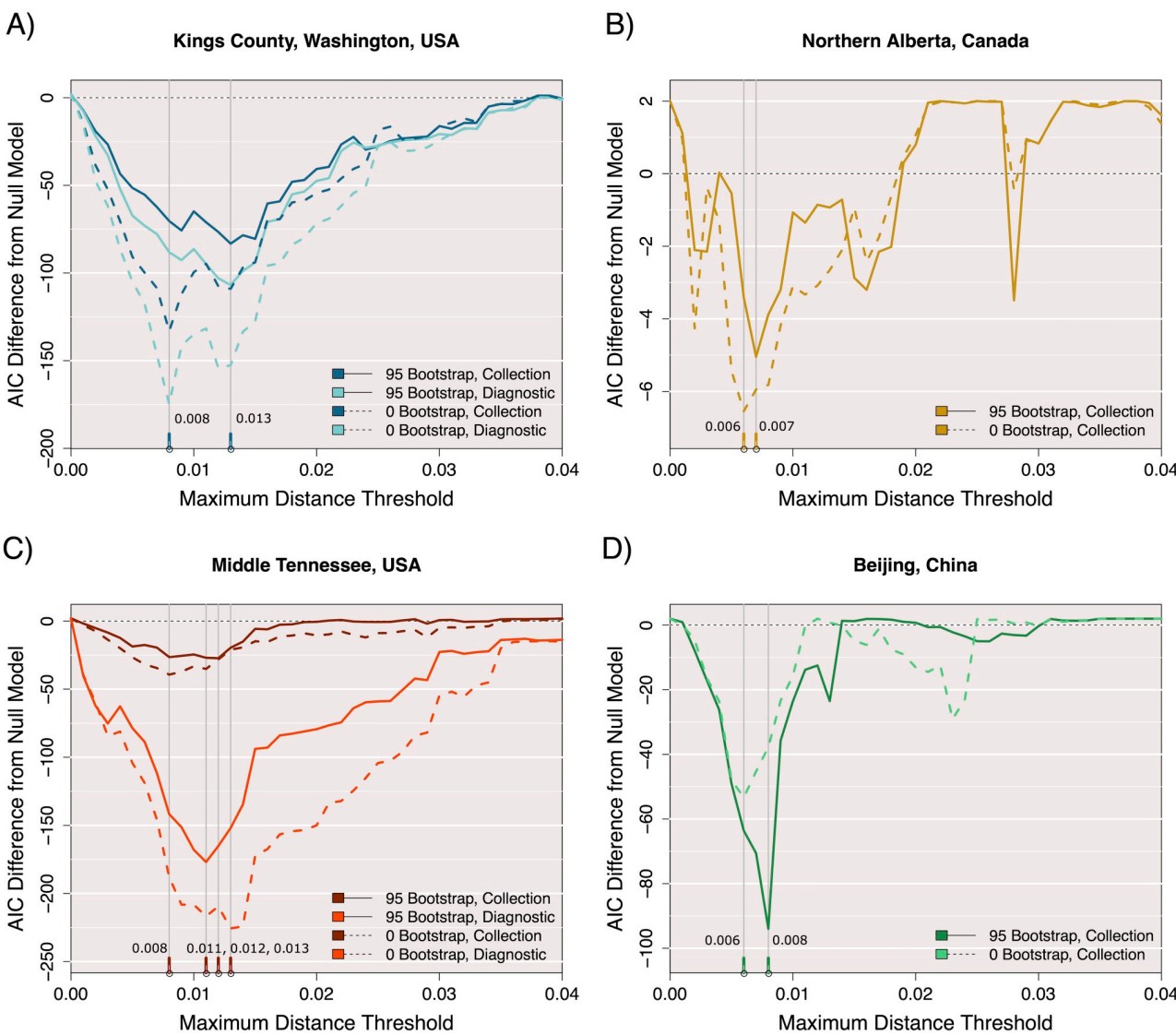

**Fig 4. Effect of bootstrap thresholds on ΔAIC profiles.** The AIC difference between two Poisson-linked models of cluster growth for all four full data sets, with clusters and growth defined at 41 different maximum distance thresholds from 0 to 0.04. The AIC of a null model where size predicts growth is subtracted from the AIC of a proposed model where size and time predict growth. For the data sets where either sequence collection date or associated patient diagnostic date could define time, both AIC difference results are shown by separate colours. Solid lines represent AIC differences obtained using an additional bootstrap threshold of 0.95 to define clusters and growth, while dashed lines were obtained without this requirement.

epidemiology literature. For example, Cluster Picker [18] defines clusters as monophyletic clades with a default bootstrap requirement of $\geq$ 95% and a maximum patristic distance between all sequences within the subtree. S5 Fig displays the ΔAIC profiles for all four data sets across 41 different patristic distance thresholds and $b_{min} = 0.95$. One notable feature of these profiles is that the ΔAIC values do not consistently converge to zero at a distance threshold of zero. The ΔAIC profiles for the Alberta and Tennessee data sets were visibly less responsive to variation in distance thresholds, making it difficult to estimate optimal thresholds.

Table 2 summarizes several clustering statistics for paraphyletic and monophyletic clustering, obtained at their respective optimal $d_{max}$ thresholds. As noted in the preceding section,

**Table 2. Cluster statistics under paraphyletic and monophyletic clustering.** 'No bootstrap' corresponds to paraphyetic clustering with $b_{min} = 0$; otherwise this threshold defaults to 95%. Optimal $d_{max}$ is the pairwise distance threshold selected by minimizing ΔAIC, in units of expected number of nucleotide substitutions. 'Number of clusters' only counts clusters with two or more background sequences, *i.e.*, this number does not include singletons. Total growth is the number of incident (new) sequences connected to clusters of background sequences. Growing clusters is the number of clusters to which incident sequences attach.

| Location | Optimal $d_{max}$ | Number of clusters > 1 | Mean cluster size | Largest cluster size | Total growth | Growing clusters |
|---|---|---|---|---|---|---|
| Washington, USA | 0.013 | 74 | 1.39 | 1657 | 68 | 30 |
| *No bootstrap* | 0.008 | 378 | 2.04 | 1560 | 73 | 36 |
| *Monophyletic* | 0.016 | 639 | 1.20 | 16 | 92 | 96 |
| Alberta, Canada | 0.007 | 104 | 1.35 | 58 | 34 | 27 |
| *No bootstrap* | 0.006 | 228 | 1.75 | 28 | 45 | 28 |
| *Monophyletic* | 0.002 | 26 | 1.04 | 4 | 52 | 64 |
| Tennessee, USA | 0.012 | 1 | 1.35 | 358 | 41 | 28 |
| *No bootstrap* | 0.008 | 621 | 1.91 | 78 | 41 | 35 |
| *Monophyletic* | 0.020 | 360 | 1.29 | 11 | 63 | 53 |
| Beijing, China | 0.008 | 135 | 1.82 | 533 | 373 | 119 |
| *No bootstrap* | 0.006 | 409 | 2.49 | 381 | 305 | 127 |
| *Monophyletic* | 0.032 | 490 | 2.13 | 46 | 613 | 275 |

removing the bootstrap threshold substantially increases the number of acceptable clusters, while the ΔAIC optima consistently shifted to slightly lower values of $d_{max}$. In addition, relaxing $b_{min}$ tends to increase the total number of incident sequences that are connected to clusters. For the Beijing data set, however, the number of incident sequences is reduced by setting $b_{min} = 0$; we attribute this to the exclusion of incident cases in paraphyletic clusters by the stricter $d_{max}$ threshold selected by ΔAIC. Under monophyletic clustering, the distribution of cluster sizes was more constrained, as it becomes increasingly likely that larger portions of the tree incorporate one or more branch lengths that causes the maximum in-cluster patristic distance to exceed the threshold.

## Validation on simulated epidemics

To determine whether clusters that were prioritized under a ΔAIC-optimized method reflected accurately the actual distribution of incident cases, we simulated an epidemic on a network using FAVITES [68] under an HIV-1 transmission model [72] that was calibrated to the heterogeneous, localized HIV-1 epidemic in British Columbia, Canada [69]. This configuration simulated the spread of HIV-1 over ten years in a population of 26,746 individuals who were connected by a preferential attachment contact network [70], starting from 3,110 infections at time 0. The transmission model assumes that infected individuals progress from acute to chronic stages of infection and become sampled upon initiating ART, which resulted in 7,748 HIV-1 *pol*-like sequences. Infections sampled in the last year of simulation ($n = 478$) were classified as incident sequences. Next, we reconstructed a phylogeny from the non-incident sequences using FastTree, and then applied our method to evaluate the growth of phylogenetic clusters by grafting incident sequences onto this tree. We obtained similar ΔAIC profiles for bootstrap thresholds of 0% and 95% with optimal distance thresholds of $d_{max} = 0.019$ and 0.018, respectively. As expected, there were substantially more numerous, smaller subtree clusters obtained with a bootstrap threshold of $b_{min} = 95\%$ ($n = 4, 217$ clusters, mean 1.7 known sequences per cluster) than with $b_{min} = 0\%$ ($n = 771$ clusters, mean 9.4 known sequences per cluster) at their respective $d_{max}$ values.

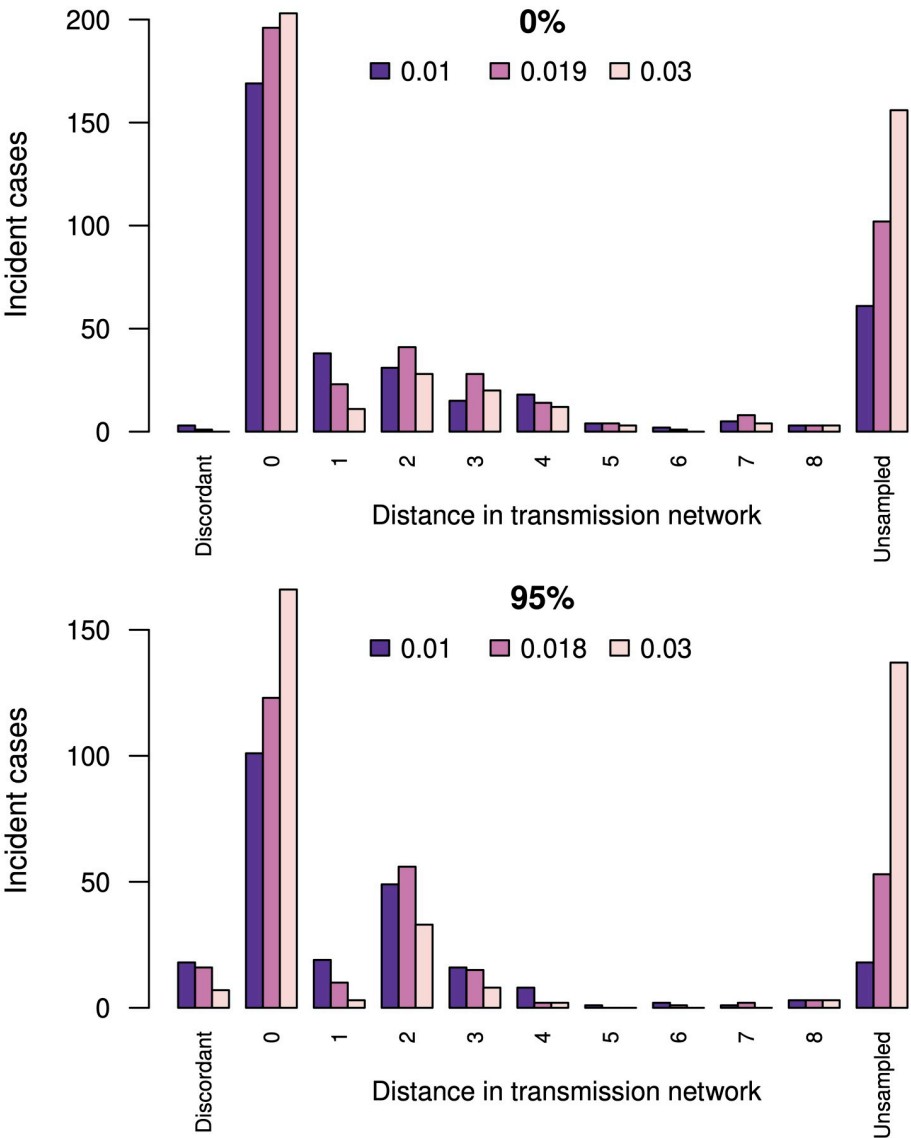

**Fig 5. Concordance between predicted growth in phylogenetic clusters and the actual (simulated) transmission network.** The top and bottom barplots summarize phylogenetic clusters obtained under bootstrap thresholds of 0% and 95%, respectively. Bars correspond to the number of incident cases mapped to phylogenetic clusters, coloured by three distance thresholds: $d_{max} = 0.01$, the ΔAIC optimum, and $d_{max} = 0.03$. *Distance in transmission network* is the shortest path in the transmission network between the incident case and any member of the predicted cluster. A distance of zero means the actual source individual is in the cluster, and distances greater than zero indicate the actual source was not sampled. *Unsampled* indicates that none of the sampled infections in the transmission history of an incident case were members of the phylogenetic cluster. *Discordant* indicates that the actual source individual was sampled but does not appear in the predicted cluster.

We retrieved the transmission events associated with the 478 incident sequences from the transmission network edge list generated by FAVITES, and then compared the locations of incident sequences among phylogenetic clusters against these 'ground truth' outputs (Fig 5). If the actual source of an incident sequence had been sampled, it was almost always a member of the predicted phylogenetic cluster; *i.e.*, 196 (99.5%) of 197 for $b_{min} = 0$ and 123 (88.5%) of 139 for $b_{min} = 95$%. On the other hand, if the actual source had not been sampled, then the shortest

distance in the transmission network from the incident sequence to any member of the predicted cluster averaged 2.88 edges for $b_{min} = 0$, and 2.46 edges for $b_{min} = 95\%$ (at their respective optimal $d_{max}$ values). These average shortest distances tended to increase with $d_{max}$, at the cost of sharply increasing numbers of incident sequences without any sampled ancestors in the cluster (labelled 'Unsampled' in Fig 5). For example, there were 102 and 53 cases—for bootstrap thresholds of 0% and 95% and the respective optimal distance thresholds—where there was no overlap between sampled infections in the transmission history of an incident sequence, and the phylogenetic cluster to which we grafted that sequence. This outcome is partly a side-effect of seeding the epidemic simulation from a large number of infections ($n = 3, 110$), which induces an equivalent number of connected components in the transmission network. Since we build a single phylogeny relating all sampled sequences, relaxing the $d_{max}$ threshold increases the chance that a subtree cluster will span multiple connected components. Since there is an unknown number of transmission events separating seed infections, we cannot quantify the distance from the incident sequence to the closest sampled infection in these cases.

## Discussion

Retrospective studies of phylogenetic clusters of HIV-1 sequences are common [18, 56, 75]. Clusters are used as a proxy for variation in transmission rates that may be statistically associated with different risk factors, such as injection drug use. In this context, defining clusters is relatively straightforward. Tracking clusters prospectively over time, however, raises significant issues for phylogenetic clustering because a cluster can be broken into a set of smaller clusters by the addition of new cases. Distance clustering methods do not suffer from this problem because pairwise distances are invariant to the addition of new data [15, 38]. This distinction is similar to the difference between single-linkage and complete-linkage clustering [35]. To circumvent this limitation of phylogenetic clustering, we have modified the conventional definition (*e.g.*, Cluster Picker [26], Tree Cluster [29]) by incorporating the concept of grafting new sequences onto a tree [54]. This allows clusters to exclude long branch lengths, making it more similar to parametric methods for phylogenetic clustering that have recently been proposed [27, 28]. An interesting challenge with this redefinition of phylogenetic clusters, however, is that relatively long terminal branch lengths can disqualify tips from ever becoming assigned to clusters. Variation in terminal branch lengths can be driven by differences in sampling/diagnosis rates among groups [45, 46]. In addition, the discordance between rates of HIV-1 evolution within and among hosts [73] can contribute to terminal branch lengths in the virus phylogeny.

In addition to clustering, phylogenetic reconstruction has also been used to approximate the transmission of an infection from one person to another. Although reconstructing transmission events at the level of individuals can confer more accurate estimates of epidemiological parameters [76], it also raises significant ethical implications due to the criminalization of HIV-1 transmission [77]. Our use of phylogenetic clustering does not resolve individual transmission events. Instead, the purpose of clustering genetically similar infections is to prospectively identify variation in transmission rates at a population level. Determining which populations are exposed to the greatest risk of onward transmission of HIV-1 in a timely manner can provide valuable information for how to allocate public health resources for outreach and prevention [5].

## Comparison to previous work

In our results, ΔAIC consistently approached zero at the extremes of the distance threshold ($d_{max}$), indicating that the addition of sample collection or diagnosis dates had no impact on our ability to predict the next cases under these extreme clustering settings. The intermediate $d_{max}$ values that minimized the ΔAIC varied among data sets and random subsamples of the same data with replacement. There was also variation when different collection date ranges were used (S4 Fig), suggesting that an optimal threshold may change over time for a given data set. This effect can be exacerbated by having a relatively small number of sequences, such as the Alberta data set in this study, or substantial variation in sample sizes per year, such as the Beijing data set. We have previously observed a similar effect for pairwise distance-based clusters [48], a clustering method that has more conventionally been used for the surveillance of molecular HIV-1 clusters [5, 37]. This observation was limited to the Washington and Tennessee data sets as the Beijing data set's sampling rate changed dramatically over time and the North Alberta data set was sampled over a relatively short time frame of 6 years. In other words, our method is sensitive to the consistency of sampling efforts over time. Selected thresholds generally fell within the range of $d_{max}$ settings used frequently in the literature— adjusting for node-to-tip versus tip-to-tip measures, this range roughly spans 0.0075 to 0.0225. For example, the original study associated with the Washington data set [56] had employed monophyletic clustering (Cluster Picker) with $b_{min} = 0.95$ and $d_{max} = 0.0075$. In contrast, our analysis favoured a more relaxed threshold for monophyletic clusters in this data set ($d_{max} = 0.016$; Table 2). However, ΔAIC was also relatively invariant to changes in $d_{max}$ under these conditions (S5 Fig).

In contrast, the effect of $b_{min}$ on phylogenetic clustering has not received as much attention as $d_{max}$, possibly due to the popularity of (non-phylogenetic) distance-based clustering methods such as HIV-TRACE [6]. Novitsky and colleagues [78] recently reported that reducing the bootstrap threshold resulted in smaller and more numerous clusters. However, they evaluated a smaller range of bootstrap thresholds ($b_{min} = 0.7 - 1.0$) and employed no distance criterion for their analysis of HIV-1 subtype C sequences from a diverse number of locations, including South Africa, Botswana and India. In our analysis, we found that ΔAIC was minimized when phylogenetic clusters were generated under no bootstrap requirement, $i.e.$, $b_{min} = 0$, which tended to yield greater numbers of small clusters with no substantial impact on selecting $d_{max}$ (Table 2). For comparison, the study originally associated with the northern Alberta data set [57] defined clusters as monophyletic clades with $b_{min} = 0.95$. Additionally, they applied a distance criterion to a time-scaled tree, such that lengths were measured in units of time, $i.e.$, $d_{max} = 5$–10 years. This was an interesting choice, because our analysis of monophyletic clustering on their data resulted in ΔAIC close to zero across a range of $d_{max}$ measured as a genetic distance (expected number of substitutions per site; S5 Fig).

Rose and colleagues [34] recently also explored the selection of clustering thresholds for phylogenetic methods in application to HIV-1. Specifically, they evaluated the ability of Cluster Picker [26] to place known HIV-1 transmission pairs ($i.e.$, epidemiologically linked heterosexual couples in the Rakai Cohort Community Study) into clusters under varying distance thresholds. They determined that $d_{max}$ between 0.04 and 0.053 were the most effective for distinguishing between epidemiologically linked and unlinked pairs in their study population. We note that their application of clustering is markedly different from our study, which focuses on the distribution of incident cases at a population level. Hence, we make no attempt to infer direct transmission between individuals, which has significant ethical and legal implications [79], and our results are unlikely to be useful for that application. Studies that focus on direct transmission require fundamentally different sampling strategies as a cluster may

represent indirect transmission through an unsampled intermediary [45, 76]. For similar reasons, any confirmed epidemiological linkages are unlikely to be useful as validation data in this framework, as the clustering is being used to quantify overall population-level differences in transmission rate, not individual-level transmission relationships.

## Limitations and future directions

There are several directions for further work. First, we have limited our optimization method to clusters extracted from a single maximum likelihood (ML) reconstruction of the phylogeny. Phylogenies can be highly uncertain, especially when substantial time has elapsed between the sampled infections and their common ancestors, or if sequences are limited to relatively short regions of the virus genomes. Thus, additional maximum likelihood trees built from the same data may yield different results. While simply repeating this workflow can provide a more stable median estimate of optimal parameters, Bayesian methods that sample multiple phylogenies from the posterior distribution defined by the confluence of the model, data and prior information are favoured for many applications of virus evolution. Since Bayesian sampling is generally limited to trees relating on the order of a hundred sequences [80], this approach is seldom used in the context of phylogenetic clustering. For example, HIV-1 population databases can comprise tens of thousands of sequences [40, 58, 81]. Some studies have overcome this limitation by applying Bayesian sampling to smaller subtrees extracted from the ML tree, but this can lead to biased estimates of transmission rates [82]. Nevertheless, it merits further investigation to assess the feasibility of computing ΔAIC profiles for a random sample of trees from the posterior distribution. Another approach would be to re-run the analysis for a number of replicate ML trees that were reconstructed under varying initial conditions, *e.g.*, random starting trees, which would make the averaged optimization results more robust to the unavoidable uncertainty in reconstructing trees.

In addition, we presently do not incorporate information about sample collection dates when reconstructing the phylogeny for cluster optimization. Since rates of evolution are often fairly consistent over time (*i.e.*, a molecular clock), these dates can be used to refine the relative positions of ancestral nodes in the tree [83], which should in turn yield a more accurate tree. Again, this information is seldom used in phylogenetic clustering studies [41]. Finally, we have limited our analysis to Poisson regression models where the variation in the number of new infections among clusters is constrained to be equal to the mean. The underlying assumption is that the rate that a new case is connected to a known case is constant, with variation in rates associated with one or more attributes of known cases *e.g.*, date of diagnosis. For instance, unexplained rate variation may cause variation in the numbers of new cases per known case to exceed the mean (*i.e.*, overdispersion). A potential means to accommodate overdispersion would be to substitute the negative binomial regression model for the Poisson model when calculating ΔAIC profiles. Our preliminary results indicate that using negative binomial regression has only a slight effect on the ΔAIC profile (S6 Fig). We also found that fitting the negative binomial model was more numerically unstable for larger data sets.

Our study has focused on the benefit of knowing the times since sample collection or diagnosis on our ability to predict the location of the next infections. Although other risk factors such as injection drug use or commercial sex work are often associated with cluster formation [21, 56, 75] and growth [15, 37, 38], sampling times are the most consistently available metadata for HIV-1 sequences. Since the Poisson regression model can hypothetically accommodate any number of predictor variables, we expect that incorporating additional metadata will drive a further decrease in ΔAIC and may shift the location of the optimal $d_{max}$. Although ΔAIC may offer a framework for variable selection in the context of cluster optimization [48],

the statistical justification for this is unclear and remains an area for further work. For example, we cannot directly compare AIC values obtained from different models at different clustering thresholds, because the underlying data have changed. Therefore, variable selection may not be as simple as determining which model minimizes the ΔAIC across thresholds. However, our results support the use of the AIC loss metric and prospective growth modeling to adjust phylogenetic clustering studies for the genetic composition of each study population.

## Supporting information

**S1 Fig. Cluster growth under varying thresholds.** The number of new cases that join any cluster (ie. the total cluster growth) plotted against the branch length threshold used to define clusters. The optimal threshold determined in Fig 4 for each data set is marked in red.
(TIF)

**S2 Fig. Area under the curve (AUC) profiles.** The AUC results for receiver operator characteristic (ROC) curves for the prediction of whether or not a cluster would acquire new cases based on recency (collection date) plotted against the distance threshold used to define clustering. The same result was also calculated without a bootstrap requirement for clustering (dashed) and with diagnostic dates used to measure recency (green). The optimal threshold determined in Fig 4 for each data set is marked in red.
(TIF)

**S3 Fig. Distribution of optimal thresholds across subsamples.** The distribution of maximum branch length thresholds which resulted in the largest difference in AIC between Poisson-linked models of cluster growth for 100 approximate likelihood trees built on 80% subsamples of the old sequences without replacement. Clusters and growth are defined at 41 different maximum distance thresholds from 0 to 0.04 with a minimum bootstrap support requirement of 95% for ancestral nodes. The AIC of a null model where size predicts growth is subtracted from the AIC of a proposed model where size and mean time predict growth.
(TIF)

**S4 Fig. Sensitivity of ΔAIC profiles to time-censored data.** Difference in AIC (ΔAIC) between Poisson-linked models of cluster growth for four separate data sets (Tennessee, USA; Washington, USA; Alberta, Canada; Beijing, China). Clusters and growth are defined at 41 different maximum distance thresholds from 0 to 0.04. The AIC of a null model where size predicts growth is subtracted from the AIC of a proposed model where size and mean time (diagnostic date for USA Data, collection dates for others) predict growth. The top row of plots used the complete data sets and each following row excludes an additional year of sample collection. ΔAIC profiles for the complete data are displayed as a dashed line on each subsequent plot for reference. The minimum value in each plot is marked with a vertical line.
(TIF)

**S5 Fig. ΔAIC profiles for monophyletic clustering.** Each line represents the difference in AIC between Poisson-linked models of cluster growth for different data sets (see inset legend). Clusters and growth are defined at 41 different maximum patristic distance thresholds within monophyletic clades from 0 to 0.04 with a minimum bootstrap support requirement of 95% for ancestral nodes. The AIC of a null model where size predicts growth is subtracted from the AIC of a proposed model where size and mean time predict growth.
(TIF)

**S6 Fig. Comparison of Poisson and negative binomial models.** Difference in AIC between null and alternative models using Poisson (black) and negative binomial (red) regressions on

cluster growth for the middle Tennessee data set. We used the *glm.nb* function in the R package MASS [84] to fit negative binomial regression models. Clusters and growth are defined at 40 different maximum patristic distance thresholds within monophyletic clades from 0.001 to 0.04 with a minimum bootstrap support requirement of 95% for ancestral nodes. The AIC of a null model where size predicts growth is subtracted from the AIC of a proposed model where size and mean time predict growth.
(TIF)

## Acknowledgments

We thank Dr. Susan Buskin and Richard Lechtenberg for their work at Public Health—Seattle & King County that made this study possible.

## Author Contributions

**Conceptualization:** Art F. Y. Poon.

**Data curation:** Yi Feng, Yuhua Ruan, Hui Xing, Joshua Herbeck, Marcia Kalish.

**Formal analysis:** Connor Chato.

**Methodology:** Connor Chato, Art F. Y. Poon.

**Software:** Connor Chato.

**Supervision:** Art F. Y. Poon.

**Validation:** Connor Chato, Art F. Y. Poon.

**Visualization:** Connor Chato, Art F. Y. Poon.

**Writing – original draft:** Connor Chato, Art F. Y. Poon.

**Writing – review & editing:** Connor Chato, Yi Feng, Yuhua Ruan, Hui Xing, Joshua Herbeck, Marcia Kalish, Art F. Y. Poon.

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
