## [Decision Letter · Decision Letter 0]

12 Apr 2022

Dear Dr. Poon,

Thank you very much for submitting your manuscript "Optimized phylogenetic clustering of HIV-1 sequence data for public health applications" for consideration at PLOS Computational Biology.

As with all papers reviewed by the journal, your manuscript was reviewed by members of the editorial board and by several independent reviewers. In light of the reviews (below this email), we would like to invite the resubmission of a significantly-revised version that takes into account the reviewers' comments.

In their paper, the authors develop a methodology for optimally clustering phylogenetic data based on the public application being taken. Their method links phylogenetic clustering with simulations models and Poisson regression to determine the appropriate threshold for calling "clusters." They then apply this method data on HIV in the USA, Canada, and China. As the authors point out, it's critical that applied methods be tied to public health objectives when the goal is mitigating and/or containing pathogen threats through public health action. While that might seem obvious, it's far too often not the case in practice. In addition, the authors also uncover a number of interesting, less-applied questions along-the-way. Again, this tight linkage between basic and applied is almost always the reality. However, I encourage the authors to pay close attention to the reviewer comments in preparing a revision. In particular, Reviewer 2's comment regarding how the sensitivity analyses were performed. In addition, I have the following comments/questions:

1.) Can the authors provide data on the trees used for these studies and/or more specific information on how to request access to the data? In addition, can you also make available all scripts used to run the analysis. In its current form, I do not believe the work is reproducible even if one were to receive access to the raw data. Addressing data availability concerns will be a criteria for publication.

2.) Given the sampling scheme? Why isn't a time-calibrated tree in BEAST more appropriate? I am not requiring the authors to make this change, but asking for--as a minimum--clarification in the text.

3.) Is there a concern about over-dispersion and have to tried fitting a negative binomial model? In my experience, this often doesn't matter, so it's ok to simply say as much in the discussion.

4.) I was surprised to see the code licensed under GPL v3 instead of MIT. Was there a reason for this decision?

We cannot make any decision about publication until we have seen the revised manuscript and your response to the reviewers' comments. Your revised manuscript is also likely to be sent to reviewers for further evaluation.

Sincerely,

Samuel V. Scarpino

Associate Editor

PLOS Computational Biology

Thomas Leitner

Deputy Editor

PLOS Computational Biology

In their paper, the authors develop a methodology for optimally clustering phylogenetic data based on the public application being taken. Their method links phylogenetic clustering with simulations models and Poisson regression to determine the appropriate threshold for calling "clusters." They then apply this method data on HIV in the USA, Canada, and China. As the authors point out, it's critical that applied methods be tied to public health objectives when the goal is mitigating and/or containing pathogen threats through public health action. While that might seem obvious, it's far too often not the case in practice. In addition, the authors also uncover a number of interesting, less-applied questions along-the-way. Again, this tight linkage between basic and applied is almost always the reality. However, I encourage the authors to pay close attention to the reviewer comments in preparing a revision. In particular, Reviewer 2's comment regarding how the sensitivity analyses were performed. In addition, I have the following comments/questions:

1.) Can the authors provide data on the trees used for these studies and/or more specific information on how to request access to the data? In addition, can you also make available all scripts used to run the analysis. In its current form, I do not believe the work is reproducible even if one were to receive access to the raw data. Addressing data availability concerns will be a criteria for publication.

2.) Given the sampling scheme? Why isn't a time-calibrated tree in BEAST more appropriate? I am not requiring the authors to make this change, but asking for--as a minimum--clarification in the text.

3.) Is there a concern about over-dispersion and have to tried fitting a negative binomial model? In my experience, this often doesn't matter, so it's ok to simply say as much in the discussion.

4.) I was surprised to see the code licensed under GPL v3 instead of MIT. Was there a reason for this decision?

Reviewer's Responses to Questions

**Comments to the Authors:**

Reviewer #1: Paper by Chato et al. entitled “Optimized phylogenetic clustering of HIV-1 sequence data for public health applications” extends the group's previous work and addresses an important question on optimizing thresholds for defining molecular HIV clusters. The main strength of the paper is that the authors propose a novel approach for calibrating phylogenetic HIV-1 clustering to specific populations based on the structure and composition of these population, which could provide a more accurate results of molecular HIV cluster analysis. The authors validate their approach using four large real-life data sets including the United States, Canada and China and provide a detailed methodology with the associated source code. This paper has a potential to advance the field and could be of a great interest to many readers.

The current paper is in a pretty good shape, “as is”, although addressing few related topics could further improve the manuscript:

- The concept of grafting new sequences into a tree is interesting and innovative, particularly for public. A comparison with a more traditional approach used currently in surveillance of molecular HIV clusters would be helpful and could serve, as a further validation.

- Some topics in Introduction, like, SARS-CoV-2, are not directly related to the study and could be omitted or shortened.

- Discussion includes a thorough literature review, which rather belongs to Introduction.

- The study limitations might need to be presented and discussed.

Minor comments:

- Full and short titles are identical.

- Figure 1 legend: The last sentence does not specify that it refers to Figure 1E; needs update.

Reviewer #2: Current phylogenetic methods commonly used to identify HIV-1 transmission clusters often require users to define patristic distance thresholds that are arbitrary and thus vary substantially across different datasets in literature. This is because these thresholds are usually chosen to best describe the transmission networks underlying different populations within which underlying epidemiological factors can differ widely alongside variations in virus sampling efforts. In this work, Chato et al. proposed a novel statistical framework to select for the optimal distance threshold. This is done by first defining paraphyletic clusters for a “background” subset of sequences using a range of plausible distance cut-off on top of satisfying a minimal bootstrap support. Next, pplacer was used to phylogenetically place “incident” sequences onto the fixed background tree and paraphyletic clustering is performed again. This step is used to simulate growth of clusters previously defined by the “background” sequences. Lastly, the inferred cluster growth are then fitted to two Poisson count models - the null model predicts growth based on cluster size while the alternative model further incorporates collection or diagnosis date of the samples as a covariate. Computing the difference in Akaike information criterion (AIC) between the two models which translates to the information gain in incorporating temporal information of the clusters, Chato et al. could then identify the patristic distance cutoff that yields the largest difference in AIC.

I have a couple of major comments that I hope the authors would find constructive and useful in improving the work:

The authors have performed sensitivity analyses whereby they randomly reduced the amount of background sequences to 80%. However, a more critical sensitivity analyses, in my opinion, is how the inferred optimal distance threshold for each geographic location would change with time. The authors currently used the most recent year of sequence collected for each of the four datasets as the “incident” subset of sequences. Can the authors perform the same analyses for a sample of historical snapshots in the past? For instance, for the Washington dataset, the authors have currently used all sequences collected before 2019 as the “background” set while 2019, the most recent year as the “incident” set. What about if the “most recent year” is now 2010 while the “background” set constitutes all data collected before 2010? How much does the distance threshold change with time in different timescales (e.g. compare between 2010 and 2019 vs between 2018 and 2019)? How much of this change then has to do with actual meaningful known difference in transmission patterns and how much of it is due simply because of variations in the amount of sequences available?

While I largely agree with the proposed framework that is fundamentally using sampling dates to systematically aid transmission cluster definition, the manuscript in its current form does not explicitly present any evaluation of how the distance thresholds inferred through this framework improve the interpretation and accuracy of the identified transmission clusters to known epidemiologically confirmed/likely transmission networks.

Minor comment:

SARS-CoV-2 transmission routes and timescales, evolutionary dynamics as well as its epidemiology are drastically distinct from HIV-1. Often times, it is difficult to identify clear transmission clusters using phylogenetic methods for SARS-CoV-2. Unless the authors are able to show clear utility of the method for identifying SARS-CoV-2 transmission clusters, I find the last sentence in line 450 misleading.

**Have the authors made all data and (if applicable) computational code underlying the findings in their manuscript fully available?**

Reviewer #1: Yes

Reviewer #2: Yes

PLOS authors have the option to publish the peer review history of their article (what does this mean?). If published, this will include your full peer review and any attached files.

Reviewer #1: No

Reviewer #2: No
---

## [Decision Letter · Decision Letter 1]

12 Aug 2022

Dear Dr. Poon,

Thank you very much for submitting your manuscript "Optimized phylogenetic clustering of HIV-1 sequence data for public health applications" for consideration at PLOS Computational Biology.

As with all papers reviewed by the journal, your manuscript was reviewed by members of the editorial board and by several independent reviewers. In light of the reviews (below this email), we would like to invite the resubmission of a significantly-revised version that takes into account the reviewers' comments.

I would like the authors to provide thorough answers to reviewer 2's remaining concerns. From my perspective, they both will require some additional analysis; however, it may be possible to sufficiently address them with appropriate additions to the discussion section.

We cannot make any decision about publication until we have seen the revised manuscript and your response to the reviewers' comments. Your revised manuscript is also likely to be sent to reviewers for further evaluation.

Sincerely,

Samuel V. Scarpino

Associate Editor

PLOS Computational Biology

Thomas Leitner

Deputy Editor

PLOS Computational Biology

I would like the authors to provide thorough answers to reviewer 2's remaining concerns. From my perspective, they both will require some additional analysis; however, it may be possible to sufficiently address them with appropriate additions to the discussion section.

Reviewer's Responses to Questions

**Comments to the Authors:**

Reviewer #1: None. All critique was addressed.

Reviewer #2: While I commend the authors' efforts in responding to my critiques and others, I am afraid I still have reservations about the work in the current form.

1. Why are only the Tennessee and Washington datasets used for the right-censoring sensitivity analyses, while the Alberta and Beijing datasets were excluded? The authors have sequence data collected over 7-10 years for Alberta and Beijing as well. In particular for Beijing, the authors observed a much broader ∆AIC curve in the 80% subsampling sensitivity analyses and that was suggested to be due to atypically higher incidence. Did they observe a similar broad curve on a year with greater incidence? If so, isn’t rapid high incidence a limitation that affects the sensitivity of this framework?

2. I am quite confused by the authors’ rationalisation of my second critique. I agree that there are ethical issues surrounding the forensic use of sequence data and phylogenetic analyses to identify HIV-1 transmission events. However, the debate surrounding this issue centres on, in my opinion and I believe the authors’, the unjust criminalisation of HIV transmissions and the ensuing demonstrable negative impacts they have on HIV public health in certain countries/communities. I disagree with the authors’ reasoning that because such data and analyses may be used to prosecute HIV infected individuals, that exempts them from validating if the clusters they had identified using their framework are accurately linked epidemiologically. Even if the authors’ framework is “designed to provide a means of prioritising clusters for public health measures by optimising the prediction of the number of new infections per cluster”, there is a need to know if the prioritised clusters are in fact accurately linked epidemiologically, and if the grafted sequences (the prospective cases) are in fact correctly placed to known clusters. This is especially important if the authors claim that their method is superior over other currently-available methods in determining the optimal clustering criteria for public health applications.

Minor

3. Please be consistent with your reference to the Washington dataset throughout the text - either stick with “Washington” or “Seattle”.

4. Could you please plot the ∆AIC curve for the full tree against that inferred for each right-censored data in Figure S3 like what you did in Figure 3?

**Have the authors made all data and (if applicable) computational code underlying the findings in their manuscript fully available?**

Reviewer #1: Yes

Reviewer #2: Yes

PLOS authors have the option to publish the peer review history of their article (what does this mean?). If published, this will include your full peer review and any attached files.

Reviewer #1: No

Reviewer #2: No
---

## [Editor Report · Decision Letter 2]

17 Nov 2022

Dear Dr. Poon,

We are pleased to inform you that your manuscript 'Optimized phylogenetic clustering of HIV-1 sequence data for public health applications' has been provisionally accepted for publication in PLOS Computational Biology.

Best regards,

Samuel V. Scarpino

Academic Editor

PLOS Computational Biology

Thomas Leitner

Section Editor

PLOS Computational Biology

---

## [Editor Report · Acceptance letter]

25 Nov 2022

PCOMPBIOL-D-22-00378R2 

Optimized phylogenetic clustering of HIV-1 sequence data for public health applications

Dear Dr Poon,

I am pleased to inform you that your manuscript has been formally accepted for publication in PLOS Computational Biology. Your manuscript is now with our production department and you will be notified of the publication date in due course.

With kind regards,

Anita Estes
